# Performance of Genotype MTBDR*sl* V2.0 over the Genotype MTBDR*sl* V1 for detection of second line drug resistance: An Indian perspective

Syed Beenish Rufai[1,2], Kulsum Umay[2], Praveen Kumar Singh[2], Sarman Singh[1,2]*

1 Division of Clinical Microbiology and Molecular Medicine, Department of Laboratory Medicine, All India Institute of Medical Sciences, New Delhi, India, 2 Department of Microbiology, All India Institute of Medical Sciences, Bhopal, India

* sarman_singh@yahoo.com, sarman.singh@gmail.com

**Data Availability Statement:** All relevant data are within the paper and its Supporting Information files.

## Abstract

Genotype MTBDR*sl* Version 1 (V1.0) was recommended as an initial test for rapid detection of pre-extensively drug resistant (pre-XDR) and extensively drug resistant tuberculosis (XDR-TB). However, in recent years a number of novel mutations are identified that confer resistance. Thus, Genotype MTBDR*sl* V2.0 was endorsed by WHO. Though, Genotype MTBDR*sl* V2.0 has been rolled out in national TB programme in 2018, there is dearth of data from India on its performance for second line drug susceptibility testing (DST). For this, performance of new version was evaluated on 113 MDR-TB isolates. The results showed that 39 (34.5%) of these isolates were resistant to FQ and 7 (6.2%) were XDR by Genotype MTBDR*sl* V2.0. Amongst the FQ resistant isolates most prevalent mutation was ΔWT3-D94G (17; 38.6%) and N538D (12; 85.7%). Among the AG/CP and KAN resistant isolates most common mutation in the *rrs* region was ΔWT1-A1401G (5; 71.4%) and C-14T (2; 28.5%) in *eis* gene. Second line Bactec MGIT-960 detected 40 (35.4%) isolates as resistant to FQ and 6 (5.3%) as XDR isolates, whereas Genotype MTBDR*sl* V1.0 also detected 39 (34.5%) as resistant to FQ but missed 2 isolates in correctly identifying as XDR (5; 4.4%). Thus, concordance of second line Bactec MGIT-960 with Genotype MTBDR*sl* V2.0 was similar (100%) for FQ detection but it has improvised the diagnostic sensitivity for correctly identifying XDR isolates. Nevertheless, the cost of Genotype MTBDR*sl* V2.0 remains an issue for screening of second line drug (SLDs) resistance from countries with high burden of MDR-TB.

## Introduction

Increasing resistance towards second line drugs (SLD's) is a major global threat leading to pre-XDR and XDR-TB which has high mortality rates due to limited treatment options and lower rates of treatment success [1,2]. Timely diagnosis and treatment remains an utmost challenge in order to stop circulating XDR-TB strains and further amplification of resistance [3–5]. Drug susceptibility testing (DST) for SLD's is gold standard for detection of drug resistance,

**Funding:** This study was supported by the grant from Indian Council of Medical Research, Government of India (5/8/5/41/2016/ECD-I) to SS.

**Competing interests:** No author has competing interest.

but varying drug potency and critical concentrations of drugs leads to poor reproducibility of DST results [6]. Among molecular approaches, line probe assays have been developed for detection of second line drug resistance like Genotype MTBDR*sl*V1.0 (Hain Lifescience GmbH Germany) which rapidly detects genotypic resistance to FQ, AG/CP and Ethambutol (EMB), within a shorter turn-around time (TAT) of 48–72 h and helps make early diagnosis of pre-XDR-TB and XDR-TB possible. Previous reports have shown very good sensitivity of this test for detection of pre-XDR and XDR TB and average sensitivity for detection of AG/CP resistance, thus in 2016 this test was recommended to be used as an initial diagnostic test along with phenotypic culture-based DST in confirmed MDR-TB cases [7–10]. With expansion of next generation sequencing methods, genes associated with resistance towards SLD's were investigated and role of *gyrB* gene for FQ resistance and *eis* gene for KAN resistance were determined [11]. Drug resistance conferring mutations in *gyrB* genes has been reported to occur in codons 500 and 538 and often co-occur with mutations of *gyrA* gene region [12]. Mutations in promotor region of *eis* gene for conferring low-level KAN mutations which is known to increase sensitivity of phenotypic KAN resistance by 9% has also been reported [13]. However, frequency of these mutations among pre-XDR and XDR-TB cases particularly from India remains unknown.

As probes for mutations associated with *gyrB* and *eis* genes were not included in Genotype MTBDR*sl* V1.0 resulting in lower sensitivity of the test. In order to overcome these issues, advanced version of Genotype MTBDR*sl* V2.0 was introduced to incorporate these mutations in *eis* promoter gene region (position -10 to -14) and *gyrB* region (codon 536 to 541)[8]. Though quite a few studies have reported the efficiency of Genotype MTBDR*sl* V2.0 for screening of pre-XDR and XDR-TB, however, more evidences are required to endorse the report especially from high TB burden countries. In 2018, Revised National Tuberculosis Control program in India has already rolled out the test for rapid detection of pre-XDR and XDR-TB however, there are scarcity of reports related to the performance of the assay. This study was thus planned in order to compare the usefulness of line probe assay (Genotype MTBDR*sl* V2.0) for detection of additional mutations in *gyrB* and *eis* genes over Genotype MTBDR*sl* V1.0 in comparison to Bactec MGIT-960 DST from India.

## Material and methods

### Study setting

A retrospective study on the mycobacterial isolates was carried out in a laboratory certified for routine diagnostics in the Department of Microbiology, All India Institute of Medical Sciences, Bhopal, India. The laboratory repository characterizes stocks and maintains all the mycobacterial isolates, obtained from routine diagnostic services for patient care. This study was performed on 113 MDR-TB isolates which were already tested for Genotype MTBDR*sl* V1.0 and Bactec MGIT-960 as published previously [7]. The ethical clearance for evaluating the performance of Genotype MTBDR*sl* V2.0 on these isolates was obtained from the institutional ethics committee of the All India Institute of Medical Sciences, Bhopal (reference # IHEC-LOP/2018/EF0102).

### Description of isolates

A total of 113 MDR-TB isolates were included in the study of which 82 (72.5%) were from pulmonary and 31 (27.4%) from extra-pulmonary sites. The sources of these isolates included sputum (76; 92.6%), Broncho-alveolar lavage (3; 3.6%), and gastric aspirate (3; 3.6%). The extra-pulmonary sources included cerebrospinal fluid (13; 11.5%), pleural fluid (6; 19.3%), pus (6; 19.3%), lymph node aspirate (3; 9.6%) and synovial fluid, urine, tissue biopsy (1; 3.2%)

each. The number of smear positive and smear negative samples were 50 (44.2%) and 63 (55.7%) in number respectively.

## Second line drug susceptibility (DST) testing using Bactec MGIT-960 system

To obtain a pure growth of MTB, 200 μL of culture suspension from the Bactec MGIT-960 tube having microscopically confirmed growth was sub-cultured on Lowenstein Jensen (LJ) medium slants and incubated at 37˚C for 21–28 days. After visible growth, a single colony was picked from LJ medium using sterile inoculating loop and inoculated in the MGIT (Mycobacteria growth indicator tube). The tube was further incubated in the Bactec MGIT-960 system until flagged positive and this growth was used for second line MGIT DST and DNA extraction for MTBDR*sl* V1.0 and MTBDR*sl* V2.0 [7,9].

The stock solution of second line drugs, OFX, AMK, KAN, and CAP (procured from St. Louis MO, USA) were dissolved, sterilized and stored at −80 ˚C in aliquots for further use as mentioned earlier [7,14]. The second line DST was performed using Bactec MGIT-960 as per the manufacturer's instructions. AST tubes were then set in the carrier rack, loaded in the Bactec MGIT-960 system and monitored repeatedly by BD Epicenter, as mentioned earlier [15].

## Line probe assay (Genotype MTBDR*sl* V1.0 and V2.0)

**DNA extraction.** DNA extraction was done as per manufacturers' instruction. Briefly, a loopful of culture isolate was suspended in 300 μl of sterile distilled water and centrifuged at 10000 x g for 15 min to pellet out sediment. Supernatant was discarded and sediment was re-suspended in 100–300 μl of sterile distilled water. This was further processed with heat lysis at 95˚C for 10 min followed by sonication and used for amplification for Genotype MTBDR*sl*V1.0 as well as for V2.0 [16].

**Amplification and hybridization.** Genotype MTBDR*sl* assays (both V1.0 as well the V2.0) were performed as per manufacturer's instructions [7,17,18]. Briefly, for amplification final volume of 50 μl of reaction mixture was used which included 35 μl of a primer-nucleotide mixture (provided along with the kit) buffer containing 2.5 mM $MgCl_2$ 1.25 U Hot Start Taq DNA polymerase (Qiagen Hilden Germany) and 5 μl of the template mycobacterial DNA [9,16–18].

**Hybridization and detection of the amplified product.** Hybridization and detection of amplified product was performed in an automated TwinCubator as per manufacturer's instructions. Briefly, denaturation of the amplification products was done by mixing 20 μL of the amplified products with 20μl of denaturing reagent (provided in the kit) for 5 min in separate troughs of a plastic well (provided in the kit). The step was followed by addition of 1 ml of pre-warmed hybridization buffer and the procedure was performed at 45˚C for 30 min followed by two steps of washing. Streptavidin conjugated with alkaline phosphatase and substrate buffer was added for colorimetric detection of hybridized amplicons. Washing was properly performed and all strips that were air dried. DNA obtained from H37RV and negative control was also tested in order to check for cross contamination during the test. Validity of the result was considered only when bands were obtained on MTB complex control (TUB) conjugate controls (CC) and amplification controls (AC) in conjunction with the target genes locus controls [7,18,19].

For Quality Control, DNA extraction was performed in the BSL-2 laboratory, master mix preparation was performed in second room with UV chamber, while PCR amplification and hybridization were performed in another room in order to avoid cross contamination.

Moreover, each batch of line probe was tested on known pan-susceptible strain H37Rv. A negative control was also added with each batch to ensure that no cross contamination occurs.

## Results

### Second line DST by Bactec MGIT-960 system

Of the 113 MDR isolates, Bactec MGIT-960 system detected 67 (59.3%) isolates as susceptible to the second line anti-TB drugs. Forty-six (40.7%) isolates were resistant to OFX while 6 (5.3%) were XDR and 3 (6.5%) were resistant to all 4 drugs OFX-KAN-AMK-CAP "Table 1".

### Results of Genotype MTBDR*s l*V1.0 and V2.0

Of the total 113 MDR-TB isolates, 69 (61.1%) isolates had no mutation in the *gyrA* and *rrs* genes. But 44 (38.9%) isolates showed resistant banding patterns (either deletion wild type band or presence of mutant band) in the *gyrA* gene region of which 39(88.6%) were FQ mono-resistant (pre-XDR). In 5 (11.4%) isolates additional *rrs* gene mutation was observed (XDR-TB).

Amongst the single codon mutations in the *gyrA* region, the most prevalent mutation was ΔWT3-D94G (17; 38.6%) followed by ΔWT2-A90V (9; 20.5%). Of the 5 (11.4%) isolates that showed resistance mutation pattern in the *rrs* region, most prevalent mutation was ΔWT1-A1401G (4; 80%) "Table 2".

In the Genotype MTBDR*sl* V2.0, no mutation was detected in the *gyrA* and *rrs* genes of 67 (59.3%) isolates. Remaining 46 (40.7%) isolates showed resistance (either deletion wild type band or presence of mutant band) singly in the *gyrA* gene region (32; 69.5%), *gyrB* gene region (1; 2.2%) or in both *gyrA* and *gyrB* gene regions (13; 28.3%). Of the 46 isolates, 39(84.8%) were FQ mono-resistant (pre-XDR) while 7 (15.2%) isolates showed additional mutations in the *rrs* and *eis* genes and these were labeled as XDR-TB isolates "Table 1".

Amongst the single codon mutations observed in the *gyrA* region, most prevalent mutation was ΔWT3-D94G (17; 38.6%) followed by ΔWT2-A90V (9; 20.5%). However, among the *gyrB* gene region, the single codon mutation (N538D) was most prevalent in (12 of 13;85.7%) isolates. Of the total 7 XDR-TB isolates, 5 (5; 71.4%) showed resistance mutation pattern in the *rrs* region (ΔWT1-A1401G); one in *eis* gene (C-14T); and another one in both *rrs* and *eis* gene (A1401G+ C-14T) "Table 3".

### Comparison of Genotype MTBDR*sl* V1.0 versus Genotype MTBDR*sl* V2.0 using MGIT-960 as standard for second line DST

The second line Bactec MGIT-960 detected total 67 (59.7%) isolates sensitive to second line drugs viz; OFX, KAN, AMK, CAP which were detected accurately by both Genotype MTBDR*sl*V1.0 and Genotype MTBDR*sl* V2.0 "Table 1". However, Genotype MTBDR*sl* V1.0 detected 2 (2.94%) isolates incorrectly sensitive to second line drugs which were detected as

**Table 1. Concordance between second line Bactec MGIT-960, Genotype MTBDR*sl* V1.0 and Genotype MTBDR*sl* V2.0.**

| Genotype MTBDRsl V2.0 (n; %age) | Genotype MTBDRsl V1.0 | | | | | Bactec MGIT-960 | | | | |
|---|---|---|---|---|---|---|---|---|---|---|
| | SEN | FQ | XDR | Agreement[a] | k coeff.[c] | SEN | FQ[b] | XDR | Agreement | k coeff. |
| SEN (67; 59.3) | 67 | 0 | 0 | 98.2 | 0.96 | 67 | 0 | 0 | 100 | 1 |
| FQ (39; 34.5) | 2 | 37 | 0 | 98.2 | 0.96 | 0 | 39 | 0 | 100 | 1 |
| XDR (7; 6.2%) | 0 | 2 | 5 | 98.2 | 0.82 | 0 | 1 | 6 | 99.1 | 0.91 |

[a] Agreement between Bactec MGIT-960, Genotype MTBDR*sl* V1.0 and Genotype MTBDR*sl* V2.0 with Bactec MGIT-960 DST using OpenEpi 3.01.

[b] FQ- Fluoroquinolone, SEN- Sensitive, XDR- Extensively drug resistant

[c] *k* coeff.- Cohens's kappa as a measure of agreement between two values.

**Table 2. Mutation pattern detected by Genotype MTBDR*sl* V1.0 assay in comparison with second line Bactec MGIT-960 on 113 MDR-TB isolates.**

| Codon mutation (*gyrA*) | Codon mutation (*rrs*) | Genotype MTBDR*sl* V1.0 | Second line Bactec MGIT-960 | Total number of isolates (%age) |
|---|---|---|---|---|
| [a]ΔWT3-D94G | ΔWT1-A1401G | FI | [b]OFX[R] [c]AMK[R], [d]CAP[R], [e]KAN[R] | 2 (4.5) |
| ΔWT3-D94G | - | [f]F | OFX[R] KAN[R] | 1 (2.3) |
| ΔWT3-D94G | - | F | OFX[R] | 14 (31.8) |
| ΔWT2-A90V | ΔWT1-A1401G | FI | OFX[R] AMK[R] KAN[R] | 1 (2.3) |
| ΔWT2-A90V | - | F | OFX[R] | 8 (18.2) |
| ΔWT3- D94A | - | F | OFX[R] | 2 (4.5) |
| ΔWT3- D94A | ΔWT1-A1401G | FI | OFX[R] AMK[R], CAP[R], KAN[R] | 1 (2.3) |
| ΔWT3- D94N/Y | - | F | OFX[R] | 2 (4.5) |
| ΔWT3- D94N/Y, D94G, D94H | - | F | OFX[R] | 1 (2.3) |
| ΔWT3- D94H | - | F | OFX[R] | 1 (2.3) |
| A90V | - | F | OFX[R] | 4 (9.1) |
| A90V, D94G | A1401G | FI | OFX[R] AMK[R] KAN[R] | 1 (2.3) |
| A90V, D94G | - | F | OFX[R] | 2 (4.5) |
| D94G | - | F | OFX[R] | 2 (4.5) |
| S91P | - | F | OFX[R] | 1(2.3) |
| A90V, D94G | - | F | OFX[R] | 1(2.3) |
| No Mutation | - | SEN | OFX[R] | 2 (4.5) |
| Total (n = 46) | | FI—5 (11.4%) | OFX[R] -40 | |
| | | F—39 (88.6%) | OFX[R] KAN[R] -1 | |
| | | SEN—2 (4.5%) | OFX[R], AMK[R], KAN[R] -2 | |
| | | | OFX[R], AMK[R], CAP[R], KAN[R] -3 | |

[a]ΔWT-Deletion of wild type band;

[b]OFX[R]-Ofloxacin resistant;

[c]AMK[R]-Amikacin resistant;

[d]CAP[R]-Capreomycin resistant.;

[e]KAN[R]-Kanamycin resistant;

[f]F-Fluoroquinolone;

[g]FI- Fluoroquinolone & Injectable (AG/CP) resistant.

OFX resistant by second line Bactec MGIT-960 as well as by Genotype MTBDR*sl* V2.0. Thus, concordance of Genotype MTBDR*sl* V2.0 with second line Bactec MGIT-960 and Genotype MTBDR*sl* V1.0 was almost perfect (*Kappa* coefficient 1 and 0.96 respectively). The phenotypic method detected 40 (86.9%) isolates as resistant to OFX, of which 38 (95%) were detected as mono resistant to FQ and 2 (6.5%) as sensitive by Genotype MTBDR*sl*V1.0. However, the Genotype MTBDR*sl* V2.0 detected 39 (97.5%) isolates as mono resistant to FQ and 1 (2.5%) isolate resistant to FQ as well as AG/CP and KAN (XDR).

The Bactec MGIT-960 phenotypic method detected total 6 (8.9%) isolates as XDR-TB of which 5 (83.3%) isolates were detected as XDR-TB and 1 (16.7%) as FQ resistant only by Genotype MTBDR*sl* V1.0. However, the genotype MTBDR*sl* V2.0 detected all 6 isolates as XDR-TB, along with 1 additional XDR-TB isolates which was detected FQ resistant by Bactec MGIT-960 as mentioned above. Thus, concordance of Genotype MTBDRsl V2.0 with Bactec MGIT-960 and Genotype MTBDR*sl* V1.0 was almost perfect (*Kappa* coefficient 0.91 and 0.82 respectively).

## Discussion

TB being a stern medical threat that affects around 10 million people worldwide [20]. In addition, the steady increase in the stretch of pre-XDR/XDR-TB over the past decade has raised

**Table 3. Mutation pattern) detected by Genotype MTBDR*sl* V2.0 assay in comparison with Bactec MGIT-960.**

| Codon mutation (*gyrA*) | Codon mutation (*gyrB*) | Codon mutation (*rrs*) | Codon mutation (*eis*) | Genotype MTBDR*sl*V2.0 | Second line Bactec MGIT-960 | Total number of isolates (%) |
|---|---|---|---|---|---|---|
| [a]ΔWT3-D94G | N538D | | | [f]F | [b]OFX[R] | 4 (8.7) |
| ΔWT3-D94G | | | C-14T | FK | OFX[R] [c]KAN[R] | 1 (2.2) |
| ΔWT3-D94G | | | | F | OFX[R] | 10 (21.7) |
| ΔWT3-D94G | | ΔWT1-A1401G | | [g]FI | OFX[R] [d]AMK[R], [e]CAP[R], KAN[R] | 2 (2.2) |
| ΔWT2-A90V | | | | F | OFX[R] | 1 (2.2) |
| ΔWT2-A90V | | | | F | OFX[R] | 8 (17.4) |
| ΔWT2-A90V | N538D | ΔWT1-A1401G | | FI | OFX[R] AMK[R] KAN[R] | 1 (2.2) |
| ΔWT3- D94A | | | | F | OFX[R] | 1 (2.2) |
| ΔWT3- D94A | N538D | | | F | OFX[R] | 1 (2.2) |
| ΔWT3- D94A | ΔWT | ΔWT1-A1401G | | FI | OFX[R] AMK[R], [e]CAP[R], KAN[R] | 1 (2.2) |
| ΔWT3- D94N/Y | N538D | | | F | OFX[R] | 1 (2.2) |
| ΔWT3- D94N/Y | | | | F | OFX[R] | 1 (2.2) |
| ΔWT3- D94N/Y, D94G, D94H | N538D | | | F | OFX[R] | 1 (2.2) |
| ΔWT3- D94H | N538D | | | F | OFX[R] | 1 (2.2) |
| A90V | | | | F | OFX[R] | 2 (4.3) |
| A90V | | | | F | OFX[R] | 2 (4.3) |
| A90V, -D94G | | | | F | OFX[R] | 2 (4.3) |
| A90V, D94G | | +WT-A1401G | | FI | OFX[R] AMK[R] KAN[R] | 1 (2.2) |
| D94G | N538D | | | F | OFX[R] | 1 (2.2) |
| D94G | ΔWT-N538D | | | F | OFX[R] | 1 (2.2) |
| S91P * | N538D | ΔWT1-1401G | +WT1-C-14T | FIK | OFX[R] | 1 (2.2) |
| A90V, D94G | | | | F | OFX[R] | 1 (2.2) |
| No mutation | N538D | | | F | OFX[R] | 1 (2.2) |
| Total (n = 46) | | | | F- 39 | OFX[R] -40 | |
| | | | | FI– 5 | OFX[R] KAN[R] -1 | |
| | | | | FK- 1 | OFX[R], AMK[R], KAN[R]-2 | |
| | | | | FIK- 1 | OFX[R] AMK[R], CAP[R], KAN[R] -3 | |

[a]ΔWT-Deletion of wild type band;

[b]OFX[R]-Ofloxacin resistant;

[c]KAN[R]-Kanamycin resistant;

[d]AMK[R]-Amikacin resistant;

[e]CAP[R]-Capreomycin resistant.;

[f]F-Fluoroquinolone;

[g]FI- Fluoroquinolone Injectable (AG/CP).

*Isolate detected as XDR by Genotype MTBDR*sl* V2.0 while it was detected as fluoroquinolone resistant by phenotypic Bactec MGIT-960.

concerns and created an alarming situation for development of reliable, accurate and rapid diagnostic tests for detection of second line drug resistance [21].

With advancement of sequencing approaches, more data has been reported to identify the mutations in the genes associated with second line drug resistance [22]. Further, advances in the nucleic acid amplification and hybridization methods, molecular tests for rapid detection of MDR-TB have been developed which have much reduced turn-around time (TAT).

However, for detection of second line drug resistance, not enough diagnostic tests are available except few *in-house* diagnostic methods and reverse hybridization assays such as Genotype MTBDR*sl* [16,18,23,24]. Yet, the second line BactecMGIT-960 remains gold standard [6].

Since the introduction of Genotype MTBDR*sl* V1.0 for determination of second line resistance in *gyrA* and *rrs* genes, WHO in 2016 recommended this test as an initial diagnostic test for confirmed MDR-TB patients for detection of FQ resistance and AG/CP class of drugs directly from smear positive clinical specimens or indirectly from culture isolates [10]. With advancement in sequencing methods, mutations conferring drug resistance were identified predominantly associated with *gyrB* regions e.g. Asn538Asp, Asp500His, Ala543Val and Thr539Asn. However these rare mutations [12,13,23]. Several low level mutations are also identified in *eis* promoter region (G-10C, G-10A, C-12T, C-14T) and *tlyA* (INS755GT) gene region conferring drug resistance to KAN and CAP [23,25–27]. However, data related to frequency of these mutations especially from high TB burden countries remains unknown or obscure [22,28]. Besides having the disadvantages for not having additional probes for *gyrB*, *eis* and *tlyA* genes, the sensitivity of Genotype MTBDR*sl* V1.0 has remained a matter of debate, ranging from 57–100% for detection of FQ and 25–100% for AG/CP drugs [19,29–35]. Though, the specificity of Genotype MTBDR*sl* V1.0 in comparison with Bactec MGIT-960 has been reported high (77–100%) [29–35].

In order to enhance the sensitivity of test a new version known as Genotype MTBDR*sl* V2.0 was introduced recently which incorporates mutations (Asn538Asp, Glu540Val) in *gyrB* and mutations in *eis* genes (G–37T, C–14T, C–12T, G–10A, C–2A) to detect low-level KAN resistance [18]. Several studies have reported performance of Genotype MTBDR*sl* V2.0 with enhances sensitivity ranging from 83.6–100% and 94.4–100% and specificity of 89.2–100% and 91.4–98.5% for FQ and AG/CP drugs, respectively [17,18,36]. Only one study from India has been reported on the efficacy of Genotype MTBDR*sl* V2.0 [36]. Hence, more studies were required in order to evaluate the performance of the test and prevalence of mutations especially in *gyrB* and *eis* genes.

This study was thus performed to evaluate the improved performance of Genotype MTBDR*sl* V2.0 over the Genotype MTBDR*sl* V1.0 while considering the phenotypic second line Bactec MGIT-960 as standard. In our study, the Genotype MTBDR*sl* V2.0 detected additionally 2 (2.94%) FQ mono-resistant isolates over Genotype MTBDR*sl* V1.0 due to presence of mutations in *gyrB* gene region which were also detected phenotypically by second line Bactec MGIT-960. Moreover, Genotype MTBDR*sl* V2.0 efficiently detected mutation in *eis* gene region in 2 (28.6%) isolates of which second line Bactec MGIT -960 detected 1 (14.3%) isolate as KAN resistant, and none of these isolates was detected by Genotype MTBDR*sl* V1.0. Thus overall, Genotype MTBDR*sl* V2.0 showed increased sensitivity for detection of FQ and XDR-TB isolates over Genotype MTBDR*sl* V1.0 and even the phenotypic second line Bactec MGIT-960.

In India the first version (V1.0) which was initially available before the launch of Genotype MTBDR*sl* V2.0 cost ranging from (1389 to 1527) US dollar each kit for 96 test, (14.4 to 15.9) US dollar each test while the cost of new version (V2.0) is approximately (1666 to 1944) US dollar each kit for 96 test (17.3 to 20.2) US dollar each test. The cost-benefit analyses for performance of Genotype MTBDR*sl* V2.0 for detection of second line drug resistance where higher rates of FQ resistance and XDR-TB is suspected should essentially be carried out whether the cost of millions of dollars, is justified or not. Because of high cost, even we wanted to test all 359 MDR isolate we had, we could test only 113 with version 2 while all were tested by version 1.0. Therefore, the national TB control programs and the manufactures must sit together and with the help of WHO they must work for reducing the prices of the kit not only for the FIND TB sites but for overall government as well as private hospitals who have setup for LPA testing services in India.

We thus conclude that Genotype MTBDR*sl* V2.0 should be used for rapid detection of second line drug resistance for FQ, AMK and KAN in order to initiate treatment among the patients. However, high cost per test of Genotype MTBDR*sl* V2.0 remains an issue. Moreover, WGS studies should be performed on large data sets to scrutinize more mutations conferring or associated with second line drug resistance in order to develop new molecular tests for rapid and accurate detection of XDR-TB.

## Supporting information

**S1 Data.**
(PDF)

## Acknowledgments

We wish to thank Mr Akash Ojha and Mr Ashish Soni for their technical help in this study.

## Author Contributions

**Conceptualization:** Sarman Singh.

**Data curation:** Syed Beenish Rufai, Kulsum Umay, Praveen Kumar Singh.

**Formal analysis:** Syed Beenish Rufai, Kulsum Umay.

**Funding acquisition:** Sarman Singh.

**Methodology:** Syed Beenish Rufai.

**Project administration:** Sarman Singh.

**Resources:** Sarman Singh.

**Software:** Kulsum Umay, Praveen Kumar Singh.

**Supervision:** Sarman Singh.

**Validation:** Syed Beenish Rufai.

**Writing – original draft:** Syed Beenish Rufai.

**Writing – review & editing:** Sarman Singh.

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
