## [Decision Letter · Decision Letter 0]

28 Nov 2019

PONE-D-19-28746

Performance of Genotype MTBDRsl V2.0 over 1 the Genotype MTBDRsl V1 for detection of second-line drug resistance: An Indian perspective

PLOS ONE

Dear Professor Singh,

Thank you for submitting your manuscript to PLOS ONE. After careful consideration, we feel that it has merit but does not fully meet PLOS ONE’s publication criteria as it currently stands. Therefore, we invite you to submit a revised version of the manuscript that addresses the points raised during the review process.

The reviewer has pointed out the need to revise the manuscript and provide more details on some specific points. One of the Tables also needs improvement

We would appreciate receiving your revised manuscript by Jan 12 2020 11:59PM. To enhance the reproducibility of your results, we recommend that if applicable you deposit your laboratory protocols in protocols.io, where a protocol can be assigned its own identifier (DOI) such that it can be cited independently in the future. For instructions see: http://journals.plos.org/plosone/s/submission-guidelines#loc-laboratory-protocols

We look forward to receiving your revised manuscript.

Kind regards,

Iddya Karunasagar

Academic Editor

PLOS ONE

Journal Requirements:

2. Please provide additional details regarding participant consent. In the ethics statement in the Methods and online submission information, please ensure that you have specified (1) whether consent was informed and (2) what type you obtained (for instance, written or verbal, and if verbal, how it was documented and witnessed). If the need for consent was waived by the ethics committee, please include this information.

We wish to thank Mr Akash Ojha and Mr Ashish Soni for their technical help in this study. This study was supported by the grant from Indian Council of Medical Research, Government of India (5/8/5/41/2016/ECD-I) to SS.

Please remove any funding-related text from the manuscript and let us know how you would like to update your Funding Statement. Currently, your Funding Statement reads as follows:  NO

Additional Editor Comments:

The reviewer has pointed out the need to revise the manuscript and provide more details on some specific points. One of the Tables also needs improvement

Reviewers' comments:

Reviewer's Responses to Questions

**Comments to the Author**

1. Is the manuscript technically sound, and do the data support the conclusions?

Reviewer #1: Yes

2. Has the statistical analysis been performed appropriately and rigorously? 

Reviewer #1: N/A

3. Have the authors made all data underlying the findings in their manuscript fully available?

Reviewer #1: Yes

4. Is the manuscript presented in an intelligible fashion and written in standard English?

Reviewer #1: Yes

5. Review Comments to the Author

Reviewer #1: I read with interest manuscript by Rufai et al reporting on the results of a head-to-head comparison of two versions on Hain MTBDRplus in an Indian TB laboratory.

In my opinion design of the study is clear, and conclusions are supported by the results. Relative lack of novelty is one of the problems but bearing in mind study was conducted in high TB incidence settings I don't consider this a major issue.

My recommendations are as follow:

- it would be helpful to know what was the principle of specimens selection, and whether it essentially should be considered a convenience sample - this is necessary to put the results in the context, and decide if these

could be considered generaliseble/expandable to other areas in India.

- I strongly recommend that Table 3 is re-formatted and band patterns are reported as combined patterns with missing WT band and specific mutation bands visible; the way it reports patterns now is extremely confusing and misleading.

- i think expanding on cost issues would be really beneficial in the context of implementation of LPAs for rapid detection of XDR/pre-XDR isolates in India and similar settings. I was really surprised to learn cost of MTBDRplus version 2.0 was significantly higher compared to V1.0 which was not the case in any settings (including high prevalence) I am familiar with. More accurate information is needed on price per test (paper reports cost per kit and it's unclear if it per 12 test box or..?). I would encourage providing more information on prices and call for urgent action from international and more importantly national bodies and considerations on the role of private providers in TB laboratory and diagnostic management in India.

6. PLOS authors have the option to publish the peer review history of their article (what does this mean?). If published, this will include your full peer review and any attached files.

Reviewer #1: No

---

## [Author Response · Author response to Decision Letter 0]

2 Jan 2020

Reviewers comments Authors Reply

1 Please ensure that your manuscript meets PLOS ONE's style requirements, including those for file naming. Thank you very much for the suggestion.

Manuscript has been modified as per PLOS ONE’s style requirements, including those file for naming. 

2 Please provide additional details regarding participant consent. In the ethics statement in the Methods and online submission information, please ensure that you have specified (1) whether consent was informed and (2) what type you obtained (for instance, written or verbal, and if verbal, how it was documented and witnessed). If the need for consent was waived by the ethics committee, please include this information. Consent was informed and written for samples received for routine TB investigations like smear microscopy, culture and DST Ethics reference number is provided (IHEC-LOP/2018/EF0102).

3. Please remove any funding-related text from the manuscript and let us know how you would like to update your Funding Statement. Currently, your Funding Statement reads as follows: NO Funding related text has been removed from the acknowledgement section and will be including in funding statement.

4. Please amend either the title on the online submission form (via Edit Submission) or the title in the manuscript so that they are identical Title on online submission and manuscript has been amended accordingly.

5. It would be helpful to know what was the principle of specimens selection, and whether it essentially should be considered a convenience sample - this is necessary to put the results in the context, and decide if these could be considered generalizable/expandable to other areas in India. The specimens selected in this study were convenience samples and included both pulmonary and extra-pulmonary samples received from different parts of India which were further tested for SIRE DST and second line DST (Rufai et al; Scientific Reports 2018). This study is an extension of the same study. The performance of genotype MTBDRsl V2.0 was evaluated over the Genotype MTBDRsl V1.0) on the aliquots of the same samples. The study has potential to be generalizable to other areas of India.

6. I strongly recommend that Table 3 is re-formatted and band patterns are reported as combined patterns with missing WT band and specific mutation bands visible; the way it reports patterns now is extremely confusing and misleading. Table 3 as well as Table 2 have been formatted as per the suggestions.

7. I think expanding on cost issues would be really beneficial in the context of implementation of LPAs for rapid detection of XDR/pre-XDR isolates in India and similar settings. I was really surprised to learn cost of MTBDRplus version 2.0 was significantly higher compared to V1.0 which was not the case in any settings (including high prevalence) I am familiar with. More accurate information is needed on price per test (paper reports cost per kit and it's unclear if it per 12 test box or.?). I would encourage providing more information on prices and call for urgent action from international and more importantly national bodies and considerations on the role of private providers in TB laboratory and diagnostic management in India. In India the first version (MTBDRsl V1.0) of LPA was available before the launch of Genotype MTBDRsl V2.0 costed ranging from (1389 to 1527) US dollar each kit for 96 tests, while the cost of new version (V2.0) is approximately (1666 to 1944) US dollar each kit of 96 tests (17.3 to 20.2) US dollar per test. The variation in rates are discounted rates based on if it is a government institutions, NGO (not for profit) or private sector.

---

## [Decision Letter · Decision Letter 1]

6 Feb 2020

Performance of Genotype MTBDRsl V2.0 over 1 the Genotype MTBDRsl V1 for detection of second-line drug resistance: An Indian perspective

PONE-D-19-28746R1

Dear Dr. Singh,

We are pleased to inform you that your manuscript has been judged scientifically suitable for publication and will be formally accepted for publication once it complies with all outstanding technical requirements.

With kind regards,

Iddya Karunasagar

Academic Editor

PLOS ONE

Additional Editor Comments (optional):

All reviewer comments have been addressed.

Reviewers' comments:

Reviewer's Responses to Questions

**Comments to the Author**

1. If the authors have adequately addressed your comments raised in a previous round of review and you feel that this manuscript is now acceptable for publication, you may indicate that here to bypass the “Comments to the Author” section, enter your conflict of interest statement in the “Confidential to Editor” section, and submit your "Accept" recommendation.

Reviewer #1: All comments have been addressed

2. Is the manuscript technically sound, and do the data support the conclusions?

Reviewer #1: Yes

3. Has the statistical analysis been performed appropriately and rigorously? 

Reviewer #1: N/A

4. Have the authors made all data underlying the findings in their manuscript fully available?

Reviewer #1: Yes

5. Is the manuscript presented in an intelligible fashion and written in standard English?

Reviewer #1: Yes

6. Review Comments to the Author

Reviewer #1: Reviewer's comments and questions have been adequately addressed so I believe it's now suitable for publication

7. PLOS authors have the option to publish the peer review history of their article (what does this mean?). If published, this will include your full peer review and any attached files.

Reviewer #1: No

---

## [Editor Report · Acceptance letter]

19 Feb 2020

PONE-D-19-28746R1 

Performance of Genotype MTBDRsl V2.0 over the Genotype MTBDRsl V1 for detection of second line drug resistance: An Indian perspective 

Dear Dr. Singh:

I am pleased to inform you that your manuscript has been deemed suitable for publication in PLOS ONE. Congratulations! Your manuscript is now with our production department. 

With kind regards,

on behalf of

Dr. Iddya Karunasagar 

Academic Editor

PLOS ONE